# High activity and selectivity of single palladium atom for oxygen hydrogenation to $H_2O_2$

Shiming Yu [1], Xing Cheng[1], Yueshuai Wang[2], Bo Xiao[1], Yiran Xing[1], Jun Ren[3], Yue Lu [2], Hongyi Li [2] ✉, Chunqiang Zhuang [2] ✉ & Ge Chen [1] ✉

Nanosized palladium (Pd)-based catalysts are widely used in the direct hydrogen peroxide ($H_2O_2$) synthesis from $H_2$ and $O_2$, while its selectivity and yield remain inferior because of the O-O bond cleavage from both the reactant $O_2$ and the produced $H_2O_2$, which is assumed to have originated from various $O_2$ adsorption configurations on the Pd nanoparticles. Herein, single Pd atom catalyst with high activity and selectivity is reported. Density functional theory calculations certify that the O-O bond breaking is significantly inhibited on the single Pd atom and the $O_2$ is easier to be activated to form *OOH, which is a key intermediate for $H_2O_2$ synthesis; in addition, $H_2O_2$ degradation is shut down. Here, we show single Pd atom catalyst displays a remarkable $H_2O_2$ yield of 115 mol/$g_{Pd}$/h and $H_2O_2$ selectivity higher than 99%; while the concentration of $H_2O_2$ reaches 1.07 wt.% in a batch.

Hydrogen peroxide ($H_2O_2$) is one of the most important chemicals in industry, used in the production of fine chemicals and medicine, rocket fuels, sterilization, bleaching, and so on[1,2]. In the conventional process, $H_2O_2$ is mainly produced via the anthraquinone method, which consists of hydrogenation and oxidation of anthraquinone successively. The quest for an ecofriendly process for $H_2O_2$ synthesis is driven by the current disadvantages including high energy consumption and heavy pollution[3,4]. Under such circumstances, the direct synthesis of $H_2O_2$ from hydrogen ($H_2$) and oxygen ($O_2$) is an efficient and clean strategy to replace the anthraquinone oxidation process[5]. However, this process is challenging because of many parallel and consecutive reactions, as shown in Fig. 1. Specifically, compared to the synthesis of $H_2O_2$, it is thermodynamically more favorable to the production of $H_2O$ via breaking O-O bonds, while the generated $H_2O_2$ also degrade through further hydrogenation and decomposition[6,7].

Palladium (Pd)[8,9] is a widely used catalyst in the direct synthesis of $H_2O_2$ due to its excellent hydrogenation activity. However, Pd is also active for side reactions and subsequent $H_2O_2$ degradation[10,11], resulting in an inferior $H_2O_2$ selectivity and poor yield. Pd-based nanoalloy catalyst (e.g., Pd-Pt, Pd-Au, Pd-Zn, Pd-Ag, Pd-Te, Pd-Sb, Pd-Sn)[5,12–23] can

effectively modify the electronic structure of Pd, thus inhibiting side reactions and $H_2O_2$ degradation. Besides, $H_2O_2$ can also be stabilized by adding strong acids or halides to the solvent, while it will cause metal shedding and needs a subsequent purification process to obtain pure $H_2O_2$[24,25]. Therefore, the rational design of a catalyst with high activity, high selectivity for oxygen hydrogenation to $H_2O_2$ as well as low degradation towards the generated $H_2O_2$ remains a formidable challenge.

The selectivity toward $H_2O_2$ is mainly determined by the competitive reactions between *OOH formation and O–O bond cleavage on catalysts which highly depend on the $O_2$ adsorption configuration[18,26–32]. The Pd nanoparticles involve various adsorption modes such as "side-on", "end-on" and "bridge", while the adsorption of $O_2$ on isolated Pd atom is usually the "end-on" type and could therefore reduce the possibility of O–O bond breaking. Thus, it would be encouraging to develop a single Pd atom catalyst to improve selectivity towards $H_2O_2$.

In this work, we have prepared a series of catalysts, among which the single Pd atom catalyst displays a remarkable $H_2O_2$ yield of 115 mol/$g_{Pd}$/h and a selectivity higher than 99%, surpassing the performance of

[1]Beijing Key Laboratory for Green Catalysis and Separation, Faculty of Environment and Life, Beijing University of Technology, Beijing 100124, P. R. China. [2]Faculty of Materials and Manufacturing, Beijing University of Technology, Beijing 100124, P. R. China. [3]North University of China, Taiyuan 030051, P. R. China. ✉e-mail: lhy06@bjut.edu.cn; chunqiang.zhuang@bjut.edu.cn; chenge@bjut.edu.cn

reported Pd-based catalysts. Besides, $H_2O_2$ degradation is also shut down, making it an ideal catalyst. The concentration of $H_2O_2$ reaches 1.07 wt.% in a batch. Density functional theory calculations reveal that the high yield and selectivity is believed to have originated from the high energy barrier of both O−O bond dissociation and $H_2O_2$ dissociation on the single Pd atom catalyst.

## Results

### Synthesis and characterization of materials

A series of $TiO_2$ (commercial P25)-supported O-Pd (oxidized Pd) catalysts were successfully synthesized through a simple hydrothermal method (Details in methods). For comparison, M-Pd (metallic Pd)/$TiO_2$ catalysts were also synthesized. Catalysts are named after the Pd loading. In the XRD pattern of O-Pd/$TiO_2$ samples (Fig. 2a and Supplementary Fig. 1a), there are no peaks associated with Pd species observed for 0.1% O-Pd/$TiO_2$ and 1% O-Pd/$TiO_2$, while a broad diffraction peak of PdO (101) around 34° appeared in 3% O-Pd/$TiO_2$ XRD

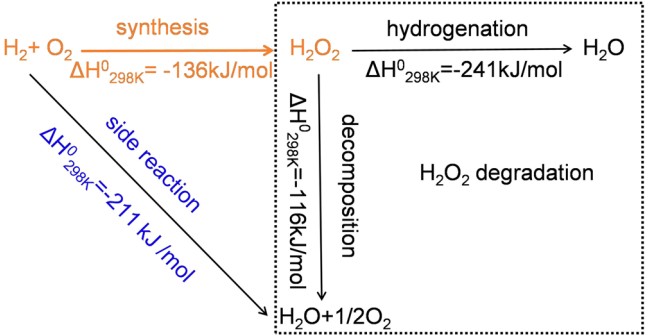

**Fig. 1 | Schematic illustration in the direct synthesis of $H_2O_2$.** all reactions in the direct synthesis of $H_2O_2$.

pattern. In contrast, the M-Pd/$TiO_2$ samples showed diffraction peaks of Pd (111) and Pd (200) around 40° and 46° except for 0.1%M-Pd /$TiO_2$ (Supplementary Fig. 1b), and the intensity of the peaks increased with the increase of palladium loading. XRD refinements results show that the lattice parameters of $TiO_2$ between samples are almost identical (Supplementary Fig. 1c–f), so the effect of different loading on the lattice of $TiO_2$ could be excluded. The catalyst morphology was further investigated by transmission electron microscopy (TEM) and aberration-corrected transmission electron microscopy (AC-TEM). No obvious Pd species can be seen from the TEM image of 0.1% O-Pd/$TiO_2$ (Supplementary Fig. 2a); however, single Pd atoms can be seen distributed over the $TiO_2$ by using AC-TEM (Fig. 2b). Because of the low $z$-contrast between Pd ($z = 46$) and Ti ($z = 22$), the single Pd atom in Fig. 2b may not be very clear. When the Pd loading reaches 1%, clusters were observed on the $TiO_2$, and the sizes are about 2 nm (Fig. 2c and Supplementary Fig. 2b). EDS mapping provided further evidence for a homogeneous distribution of Pd over the catalyst (Fig. 2d).

X-ray absorption spectroscopy (XAS) measurements were performed to probe the local environment of Pd atoms in O-Pd/$TiO_2$ and M-Pd/$TiO_2$. The X-ray absorption near edge structure (XANES) region of the XAS spectrum provides information about the oxidation state of Pd. The Pd K-edge absorption edge position for O-Pd/$TiO_2$ were like that of standard PdO sample (Fig. 3a), they were at higher photon energies than Pd metal, indicating that the Pd atoms in O-Pd/$TiO_2$ were in the oxidation state. Fourier transformed R-space curves of the Pd K-edge EXAFS spectra revealed the bonding environment of Pd species in O-Pd/$TiO_2$, an obvious peak at 1.75 Å was observed in the R-space spectrum of samples which was believed to be a Pd-O bond (Fig. 3b). The Pd metal foil and 1% M-Pd/$TiO_2$ showed an intense Pd-Pd feature at around 2.5 Å, which was absent in the O-Pd/$TiO_2$ samples (Fig. 3b), confirming that no Pd-Pd bonds were present in the O-Pd/$TiO_2$. And the fitting results showed the coordination number of Pd-O is 4 for 0.1% O-Pd/$TiO_2$ (Fig. 3c and

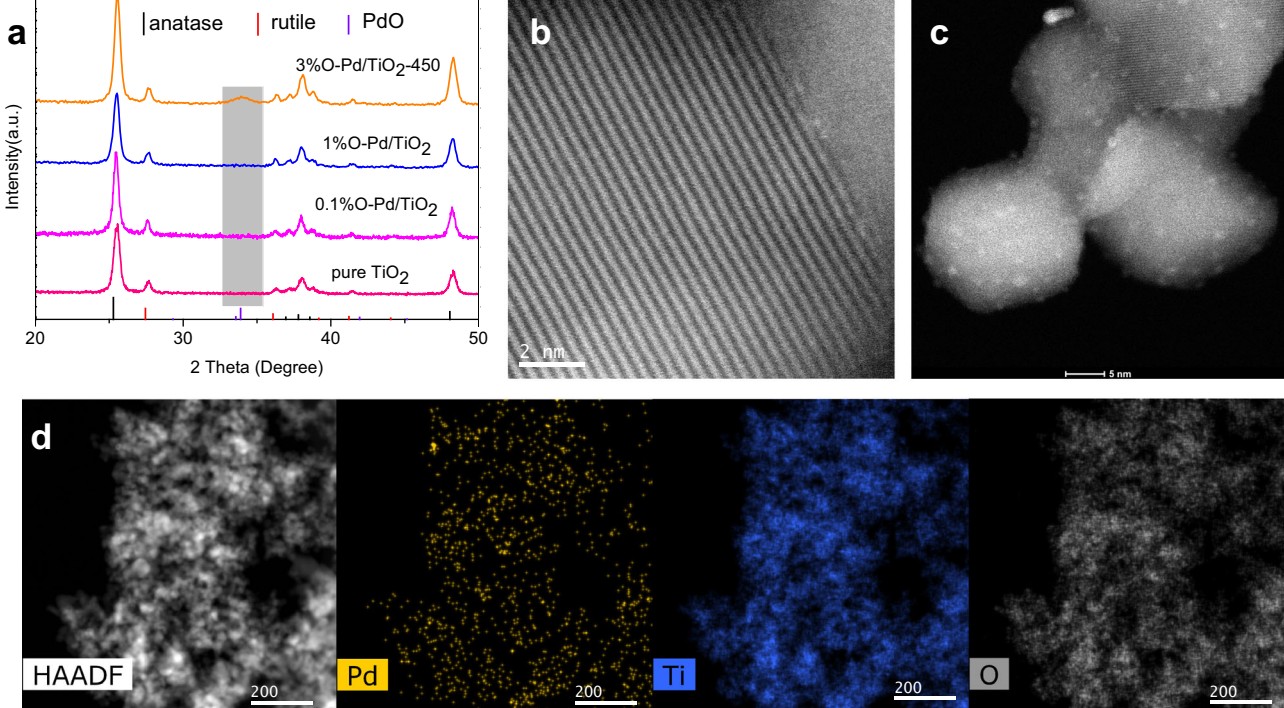

**Fig. 2 | Structure characterization of the catalysts. a** XRD patterns of O-Pd/$TiO_2$. **b** HAADF-STEM images of 0.1%O-Pd/$TiO_2$ with 2 nm scale bars. **c** HAADF-STEM images of 1% O-Pd/$TiO_2$ with 5 nm scale bars. **d** STEM-EDS elemental mappings 0.1% O-Pd/$TiO_2$, scale bars 200 nm.

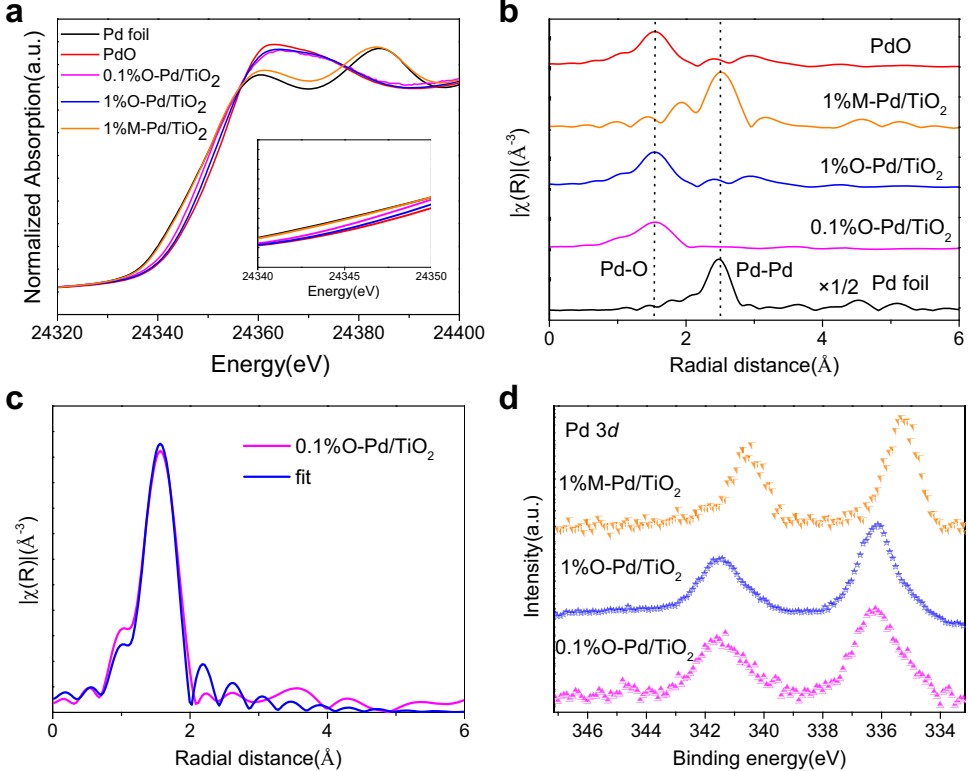

**Fig. 3 | Spectroscopic characterization. a** XANES spectra at the Pd K-edge and **b** The Fourier transforms of Pd K-edge EXAFS spectra for catalysts, PdO, and Pd foil.

**c** The FT EXAFS fitting spectrum of 0.1% O-Pd/TiO$_2$ at R-space. **d** The X-ray photo-toelectron spectra of catalysts.

Supplementary Table 1), representative of the oxidized Pd single atoms on the surface of TiO$_2$[33,34]. These results confirmed that the Pd was atomically dispersed on TiO$_2$, which was consistent with the results of AC-TEM. The Pd K-edge absorption edge position and Fourier transformed R-space curves for M-Pd/TiO$_2$ were like Pd foil, indicating that the metallic Pd states in M-Pd/TiO$_2$ samples (Fig. 3a and Supplementary Fig. 3). X-ray photoelectron spectroscopy (XPS) analyses were presented in Fig. 3d, there is a strong signal of Pd$^{2+}$ species (336.3 eV) in the spectra of 0.1% and 1% O-Pd/TiO$_2$. For M-Pd/TiO$_2$, we find that there is a strong signal of Pd$^0$ (334.9 eV) (Fig. 3d and Supplementary Fig. 4a). The absence of Cl peaks in the XPS spectra indicates that most of the chloride ions have been removed during the washing process, so the influence of chlorine ions on catalytic performance is excluded. (Supplementary Fig. 4b, c).

**Catalytic performance.** The O$_2$ hydrogenation to H$_2$O$_2$ was performed in a batch reactor with temperature at 2 °C controlled by an ice bath. We show that the catalytic performance of Pd catalysts in oxidation state (O-Pd/TiO$_2$) (including single Pd atom (0.1%), Pd cluster (1%)) are all better than that of metallic Pd catalyst (M-Pd/TiO$_2$). (Table 1, Entry 1–11). Particularly, the single Pd atom catalytic performance was up to 115 mol/kg$_{cat}$/h, and the selectivity of H$_2$O$_2$ was more than 99% (Table 1, Entry 1–2), which is superior to all other samples as well as the reported state-of-the-art catalysts (Table 1, Entry 1–2, 3–20). The Pd cluster catalyst (1% O-Pd/TiO$_2$) has a H$_2$O$_2$ yield of 79 mol/kg$_{cat}$/h, while the selectivity is 58% (Table 1, Entry 3–5), and the PdO nanoparticles demonstrate only a 7% selectivity to H$_2$O$_2$. For M-Pd/TiO$_2$, the H$_2$O$_2$ yield are much lower than that of O-Pd/TiO$_2$ (Table 1, Entry 8-11). For example, the H$_2$O$_2$ yield of 3% and 5% M-Pd/TiO$_2$ were 46 and 42 mol/kg$_{cat}$/h, respectively. Interestingly, the catalytic activity of 0.1% M-Pd/TiO$_2$ is very limited (Table 1, Entry8), as H$_2$O$_2$ yield was not detected. Also, we have synthesized 9% Pd/C catalyst, and the activity

and selectivity of Pd/C catalyst was inferior to both O-Pd/TiO$_2$ and M-Pd/TiO$_2$ catalysts. (Table 1, Entry 12).

Although the performance of the O-Pd/TiO$_2$ catalyst is superior to that of the M-Pd/TiO$_2$, whether the oxidized Pd species possessing high activity and selectivity remain controversial. The Burch et al. propose that M-Pd species has a higher activity and selectivity to H$_2$O$_2$[8]. Conversely, the Choudhary and Gaikwad et al. showed that oxidized Pd species exhibit better catalytic activity and selectivity by loading M-Pd and PdO on different oxide supports[25,35]. H$_2$ is easily activated on Pd$^0$ sites, while O$_2$ tends to be dissociated on successive Pd$^0$ sites; however, O$_2$ is stable on the surface of PdO[26]. Thus, both Pd$^0$ and Pd$^{2+}$ species play vital roles in H$_2$O$_2$ synthesis. Recently, the DFT calculation by Wang et al. suggested that PdO (101) can effectively inhibit the dissociation of O−O bonds, leading to better activity and selectivity than Pd (111)[27], which is consistent with our experimental results.

The influence of reaction conditions (catalyst feeding and reaction time) on catalytic performance were further studied. For 0.1%O-Pd/TiO$_2$, the amount of H$_2$O$_2$ increases with the increase of catalyst feeding. For example, it can produce 144 µmol of H$_2$O$_2$ in half an hour when 2.5 mg 0.1%O-Pd/TiO$_2$ catalysts are fed, 10 mg can produce 390 µmol of H$_2$O$_2$ (78 mol/kg$_{cat}$/h), corresponding to 0.15% concentration (Fig. 4a), H$_2$ conversion is 10.43%, which is also comparable to PdSn catalysts (9% of H$_2$ conversion, 61 mol/kg$_{cat}$/h, 96% of selectivity) (Table 1, Entry 17). It is also noteworthy that the H$_2$O$_2$ selectivity of 0.1%O-Pd/TiO$_2$ is >99% regardless of the quantity of 2.5 mg, 5 mg, or 10 mg (Fig. 4b). On the contrary, for clusters and nanoparticles, the increase in the amount of H$_2$O$_2$ production is not obvious, but their H$_2$O$_2$ selectivity gradually decreases (Fig. 4a, b). Reaction time was extended from half an hour to three hours. We found that when the reaction time reached 2.5 h, the production of H$_2$O$_2$ was up to 1877 µmol (0.75% concentration) for 0.1%O-Pd/TiO$_2$ (Fig. 4c). However, when the reaction time is more than 2.5 h, the concentration of H$_2$O$_2$

**Table 1 | Comparison of performance of the representative catalysts with our catalysts[a]**

| Entry | Catalyst | Pd loading (ICP-AES) | $H_2O_2$ yield (mol/kg$_{cat}$/h) | $H_2O_2$ yield (mol/g$_{Pd}$/h) | $H_2O_2$ degradation (mol/kg$_{cat}$/h) | $H_2$ conversion (100%) | $H_2O_2$ Selectivity (100%) | Ref. |
|---|---|---|---|---|---|---|---|---|
| **Our results** | | | | | | | | |
| Oxidized | | | | | | | | |
| 1 | 0.05%O-Pd/$TiO_2$ | 0.05% | 54 | 108 | n.d. | 1.84 | > 99 | – |
| 2 | 0.1%O-Pd/$TiO_2$ | 0.1% | 115 | 115 | n.d. | 3.85 | > 99 | – |
| 3 | 0.5%O-Pd/$TiO_2$ | 0.5% | 50 | 10 | n.d. | 4.31 | 40 | – |
| 4 | 1%O-Pd/$TiO_2$ | 1% | 79 | 7.9 | n.d. | 4.70 | 58 | – |
| 5 | 2%O-Pd/$TiO_2$ | 1.5% | 79 | 5.26 | n.d. | 5.44 | 50 | – |
| 6 | 3%O-Pd/$TiO_2$–450 | 2.7% | 49 | 1.81 | n.d. | 7.67 | 22 | – |
| 7 | 3%O-Pd/ $TiO_2$–600 | 2.7% | 23 | 0.85 | n.d. | 11.32 | 7 | – |
| Metallic | | | | | | | | |
| 8 | 0.1%M-Pd/$TiO_2$ | 0.1% | n.d | n.d. | n.d. | n.d. | n.d. | |
| 9 | 1%M-Pd/$TiO_2$ | 0.8% | 61 | 7.62 | 518 | 4.89 | 43 | – |
| 10 | 3%M-Pd/$TiO_2$ | 2.7% | 46 | 1.70 | 714 | 6.60 | 24 | – |
| 11 | 5%M-Pd/$TiO_2$ | 4.6% | 42 | 0.91 | 867 | 10.33 | 14 | – |
| 12 | 9%Pd/C | 9% | 30 | 0.33 | 1054 | 25.83 | 4 | – |
| **Results in the literature** | | | | | | | | |
| 13 | $Pd_5Zn$/$Al_2O_3$ | 0.85% | 216 | 25.431 | 177.54 | 56.6 | 78.5 | [17] |
| 14 | 0.5%Au/0.5% Pd/$TiO_2$ | 0.5% | 99 | 19.8 | 230 | – | 70 | |
| 15 | $Pd_6Pb$ NRs/$TiO_2$-H-A | 3.16% | 170.1 | 5.667 | 260 | 40 | 56.7 | [5] |
| 16 | 2.5%Au-2.5%Pd/C | 2.5% | 110 | 4.4 | n.d. | – | 80 | [4] |
| 17 | 3 wt%Pd–2 wt% Sn/$TiO_2$ | 3% | 61 | 2.03 | n.d. | 9 | 96 | [23] |
| 18 | 5 wt%Pd@NiO-3/$TiO_2$ | 5% | 89 | 1.78 | 8 | – | 91 | [16] |
| 19 | 2.5%Au–2.5%Pd/C | 2.5% | 160 | 6.4 | n.d. | – | >98 | [22] |
| 20 | $Pd_1Au_{220}$ | 0.011% | – | – | – | – | 95 ± 3 | [30] |

[a]$H_2O_2$ yield was determined under the following reaction conditions: 5% $H_2$/$CO_2$ (3.0 MPa) and 25% $O_2$/$CO_2$ (1.2 MPa), 8.5 g solvent (2.9 g water, 5.6 g $CH_3OH$), 2.5 mg catalyst, 2 °C, 1200 rpm, 30 min. $H_2O_2$ degradation was under standard reaction conditions: 5% $H_2$/$CO_2$ (3.0 MPa), 8.5 g solvent (5.6 g $CH_3OH$, 2.34 g $H_2O$, and 0.56 g 30% $H_2O_2$), 2.5 mg catalyst, 2 °C, 1200 rpm, 30 min. n. d., not detected. To ensure the reliability of the data, all the above experiments have to be tested for nine times, the data presented was the average value, the error of $H_2O_2$ yield and selectivity are within 1% and 4% respectively.

remains at 0.75%. The explanation might be that the large gas consumption in the reactor hinder the further generation of $H_2O_2$. To verify this point, we renew the gas in the reactor after the reaction of 2.5 h, and proceed with the reaction for the following 2.5 h (note the remaining gas in the reactor was completely discharged to 0 Mpa and then injected with 3.0 Mpa 5%$H_2$/$CO_2$ and 1.2 Mpa 25% $O_2$/$CO_2$). The results show that the concentration of $H_2O_2$ rose from 0.75% to 1.07% (2685 µmol). In general, $H_2O_2$ selectivity will decrease because of the side reactions and $H_2O_2$ degradation in a long-term reaction[8,23,24,31,32]. Interestingly, we found that the selectivity of 0.1%O-Pd/$TiO_2$ is always >99% no matter the reaction time (Fig. 4d). But for clusters and nanoparticles, the $H_2O_2$ selectivity does decline (Fig. 4d). One interpretation of this phenomenon is that as $H_2$ conversion increases, selectivity decreases due to $H_2O_2$ degradation. This can be better understood by comparing selectivity as a function of conversion for the different catalysts (Supplementary Fig. 5).

Therefore, we further measured the degradation rate of $H_2O_2$ under similar reaction conditions (3.0 Mpa 5%$H_2$/$CO_2$, the initial concentration of $H_2O_2$ is 2 wt.%). Firstly, the experiment shows that $H_2O_2$ does not degrade on $TiO_2$ support itself at 2 °C (Supplementary Table 2), which is consistent with the results previously reported by Edwards et al.[22]. Interestingly, the $H_2O_2$ degradation rate was not detected on the O-Pd/$TiO_2$ catalysts, but there was a significant $H_2O_2$ degradation rate on the M-Pd/$TiO_2$ catalysts; with the increase of metallic Pd loading, the degradation rate of $H_2O_2$ becomes higher

(Table 1, Supplementary Table 3). For example, the $H_2O_2$ degradation rate of 1% and 5%M-Pd/$TiO_2$ were 518 and 867 mol/kg$_{cat}$/h. To further understand the $H_2O_2$ degradation performance of catalysts, we have conducted a series of $H_2O_2$ degradation experiments under different conditions of atmosphere (5% $H_2$/$CO_2$ (3.0 MPa), 25% $O_2$/$CO_2$ (3.0 MPa) and pure $N_2$ (3.0 MPa), 5%$H_2$/$N_2$ (3.0 Mpa). (Supplementary Table 4).

Under the 5% $H_2$/$CO_2$ (3.0 MPa), the degradation of $H_2O_2$ was not detected on O-Pd/$TiO_2$ (Supplementary Table 4, Entry 1); to eliminate the influence of $CO_2$, we used 5% $H_2$/$N_2$ for the degradation experiment, the degradation rate was still not detected (Supplementary Table 4, Entry 8); however, the amount of $H_2O_2$ gradually decreased with time for M-Pd/$TiO_2$ catalyst. (Fig. 4e; Supplementary Table 4, Entry 2–3). On the other hand, we have not detected any degradation for both O-Pd/$TiO_2$ and M-Pd/$TiO_2$ catalyst under 25% $O_2$/$CO_2$ (3.0 MPa) or pure $N_2$ (3 Mpa). (Supplementary Table 4, Entry 4–7), which indicates the slow degradation of $H_2O_2$ in the absence of $H_2$. These observations suggested that the degradation rate of $H_2O_2$ is extremely low on the O-Pd/$TiO_2$ catalyst even in the presence of $H_2$. The result is meaningful for the degradation of $H_2O_2$ can be avoided without adding any inhibitors or pretreatment/post-treatment of catalysts.

Finally, we have investigated the stability of 0.1% O-Pd/$TiO_2$ catalyst (Fig. 4f) by measuring the $H_2O_2$ yield in the cycle test. The $H_2O_2$ yield decreased from 115 to 74 mol/kg$_{cat}$/h after reused five times. In the HAADF-STEM image of the used 0.1% O-Pd/$TiO_2$ (Supplementary

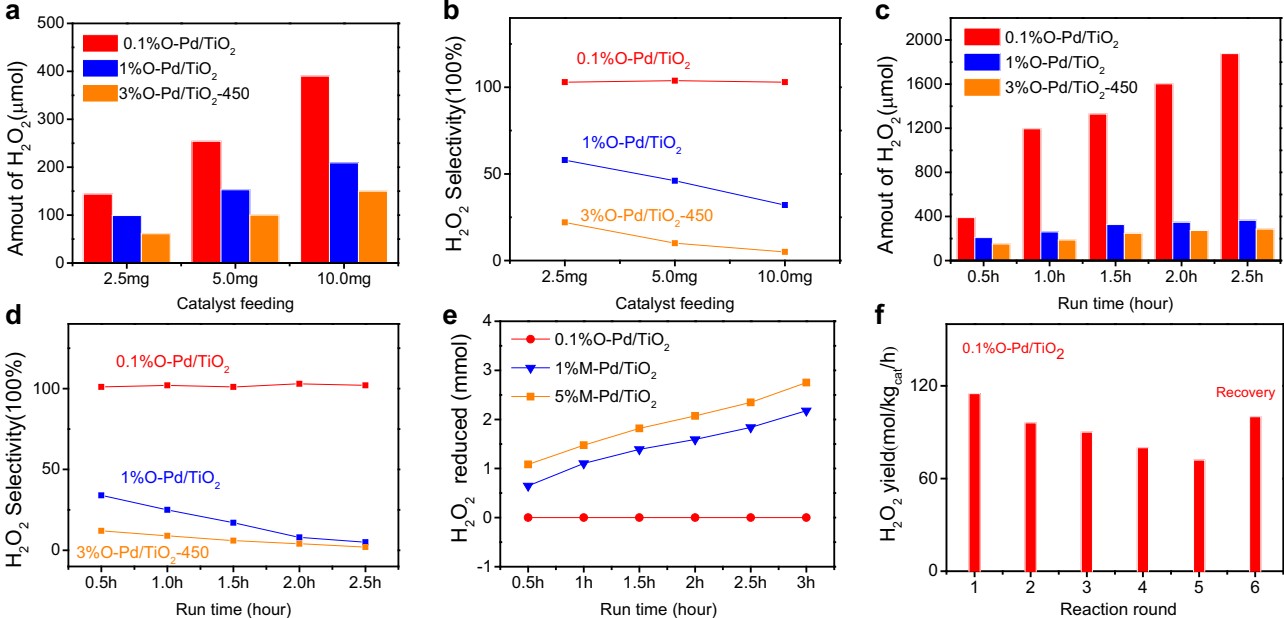

**Fig. 4 | Catalytic performance. a**, **b** The amounts of $H_2O_2$ and the $H_2O_2$ selectivity of different catalysts feeding. Reaction time: 30 min. **c**, **d** The amounts of $H_2O_2$ and the $H_2O_2$ selectivity of different reaction time. Catalyst feeding: 10 mg. **e** $H_2O_2$ degradation test. Reaction conditions: 5%$H_2$/$CO_2$ (3.0 Mpa), 2.5 mg of catalyst, 8.5 g solvent (2% $H_2O_2$),1200 rpm, 2 °C. **f** Stability test. Reaction conditions: 5%$H_2$/$CO_2$ (3.0 Mpa), 25%$O_2$/$CO_2$ (1.2 Mpa), 2.5 mg of catalyst, 8.5 g solvent, 1200 rpm, 2 °C, 30 min.

Fig. 6), we have observed some aggregated Pd species (about 3 nm Pd nanoparticle) on the $TiO_2$ particle by using EDS mapping, however, the homogenous distribution of Pd species in other places suggest that many single Pd atom still existed. The XPS results of the used catalyst reveal the existence of oxidized Pd species other than the metallic Pd species (Supplementary Figs. 7 and 8), which is consistent with the STEM result. Since the metallic Pd species showed very poor $H_2O_2$ yield in the experiment (Table 1, Entry 9–11), the observed Pd aggregation might mainly account for the loss of $H_2O_2$ yield in the recycle test. The decrease of $H_2O_2$ yield in the recycle experiment further indicates that the performance of the catalyst after reaction (co-existence of a single Pd atom and Pd nanoparticles) is not as good as that of the fresh catalyst, highlighting the important role of the initial single Pd (2+) state.

To further confirm this hypothesis, we have heated the used catalyst (after 5 cycles) at 350 °C for 3 h in the air, and the $H_2O_2$ yield increase from 74 mol/kg$_{cat}$/h to 100 mol/kg$_{cat}$/h, which is close to the fresh catalyst (Fig. 4f, round 6). It was also reported that the anneal treatment in the oxidative atmosphere will re-disperse the noble nanoparticle to a single noble metal atom on the oxide support[36–38]. Thus, it's reasonable to believe that the recovered $H_2O_2$ yield might be caused by the recovery of a single $Pd^{2+}$ atom through annealing in the air, which also suggests the anneal treatment in the air is an effective method to refresh the used catalyst. The result again confirms that the single $Pd^{2+}$ atom accounts for the high activity and selectivity towards the direct synthesis of $H_2O_2$.

### Density functional theory calculations

As mentioned in the introduction section, the $H_2O_2$ selectivity is mainly determined by the competitive reactions between *OOH formation and O−O bond cleavage on catalysts. For these reasons, first-principles calculations are used to investigate the dissociation of $O_2$ and $H_2O_2$ over single atom and PdO clusters. Single Pd atom ($Pd_1$/$TiO_2$) and PdO clusters ($Pd_8O_8$/$TiO_2$) structure models were optimized on the surface of $TiO_2$ (101) (Supplementary Fig. 9). The $Pd_1$/$TiO_2$ model is based on the palladium-oxygen coordination number fitted by EXAFS, and $Pd_8O_8$ is extracted from the bulk phase of PdO, which simulates the

coordination information of Pd and O in large nanoparticles. It is shown that $O_2$ is adsorbed by "Pd-O-O" configuration on $Pd_1$/$TiO_2$ and is mainly adsorbed by "Pd-O-O-Pd" configuration on $Pd_8O_8$/$TiO_2$ (Supplementary Fig. 10). The oxidation state of Pd single atom estimated by bader charge analysis is (+1.73) between +1 and +2 (Supplementary Fig. 11), which is consistent with the XAS results.

Firstly, we have calculated the dissociation energy barrier of $O_2$ on $Pd_1$/$TiO_2$ and $Pd_8O_8$/$TiO_2$, respectively. The dissociation of $O_2$ on $Pd_1$/$TiO_2$ is a high energy process that is endothermic by 1.89 eV (Fig. 5a), while the dissociation energy barrier of $O_2$ on $Pd_8O_8$/$TiO_2$ needs only 1.08 eV, indicating that the dissociation of $O_2$ on $Pd_1$/$TiO_2$ is not favorable. And compared with the reported results of related DFT work (Supplementary Table 5), we find that the dissociation energy barrier of O−O bond on $Pd_1$/$TiO_2$ is the highest. This difference in the dissociation barrier can be attributed to structural differences (i.e., the adsorption configuration of oxygen on Pd). Oxygen can only be dissociated by migrating one oxygen atom to the nearby Ti site on $Pd_1$/$TiO_2$, whereas on $Pd_8O_8$/$TiO_2$ it can be directly fractured by "Pd-O-O-Pd" (Supplementary Fig. 12). Moreover, we found that $H_2$ is more easily activated on $Pd_1$/$TiO_2$ than $Pd_8O_8$/$TiO_2$ (Fig. 5b).

Similarly, the formation of *OOH on $Pd_1$/$TiO_2$ only needs to overcome the energy barrier of 0.81 eV, while $Pd_8O_8$/$TiO_2$ needs to overcome 1.25 eV (Fig. 5c). Structurally, the reason is that the adsorbed oxygen can directly capture the adsorbed hydrogen at the single Pd atom site, while clusters need to capture hydrogen from other Pd sites (Supplementary Fig. 13).

In addition, the reaction barriers for the step of OOH* hydrogenation to $H_2O_2$ and OOH* dissociation were investigated. On $Pd_1$/$TiO_2$, the formation of $H_2O_2$ only needs to cross an energy barrier of 0.16 eV, while OOH* dissociation needs to overcome 0.19 eV of barrier (Fig. 5d). In contrast, the formation of $H_2O_2$ needs to overcome 0.27 eV on $Pd_8O_8$/$TiO_2$, OOH* dissociation needs to overcome 0.35 eV of barrier (Fig. 5e). The calculated results suggest that $Pd_1$/$TiO_2$ is much more favorable than $Pd_8O_8$/$TiO_2$ for $H_2O_2$ formation.

The $H_2O_2$ degradation involved two steps ($H_2O_2 \rightarrow 2OH^*$; $OH^* + H^* \rightarrow H_2O$). The calculated energy barrier of $H_2O_2$ dissociation into $2OH^*$ on $Pd_1$/$TiO_2$ (0.58 eV) is higher than that of $Pd_8O_8$/$TiO_2$ (0.07 eV)

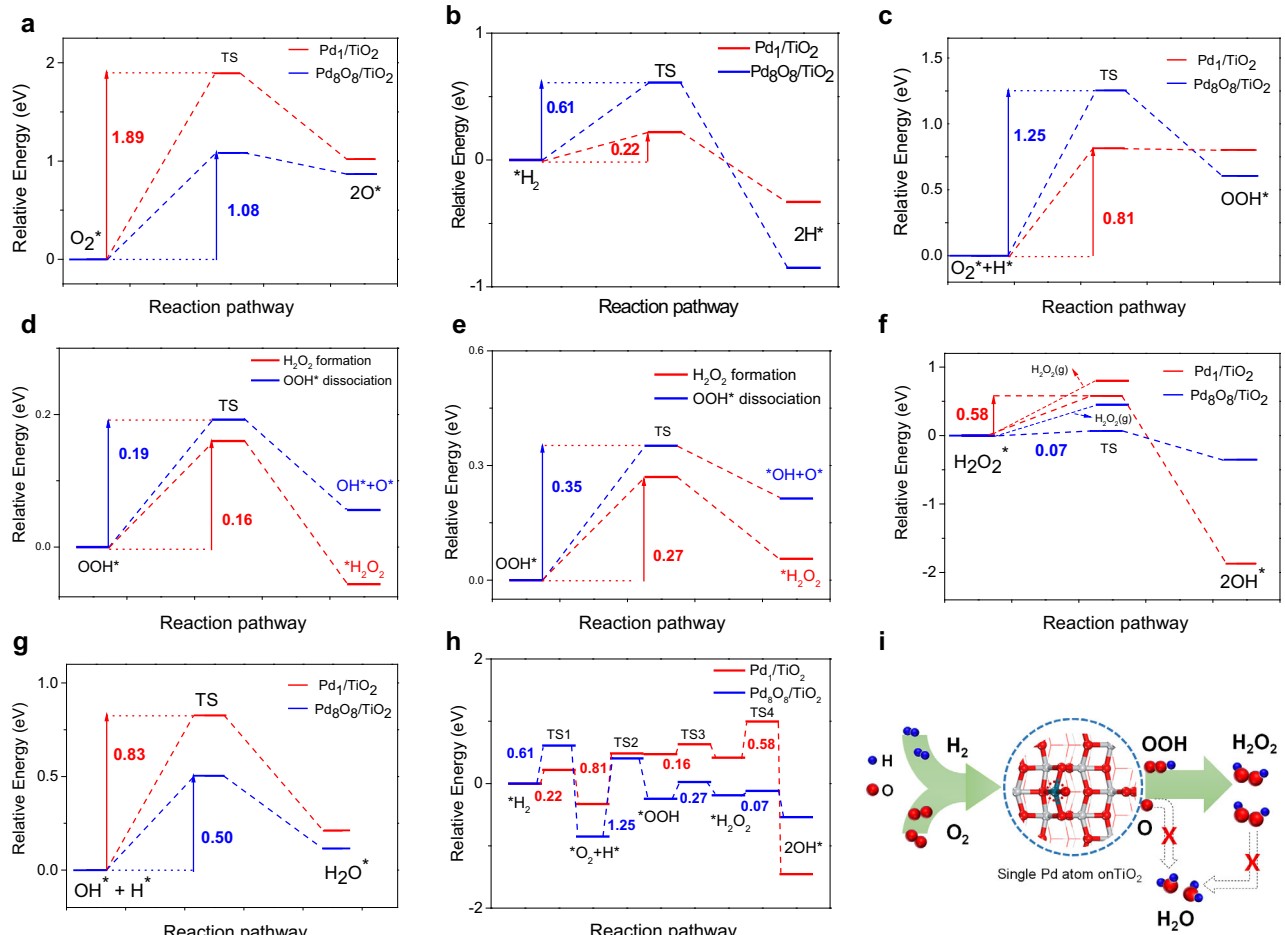

**Fig. 5 | DFT calculations. a** The reaction energy barriers of adsorbed $*O_2$ dissociation steps on $Pd_1/TiO_2$ and $Pd_8O_8/TiO_2$. **b** Energy profiles for $H_2$ dissociation on $Pd_1/TiO_2$ and $Pd_8O_8/TiO_2$. **c** The reaction energy barriers of adsorbed $*O_2$ hydrogenation steps on $Pd_1/TiO_2$ and $Pd_8O_8/TiO_2$. **d** The reaction energy barriers of $H_2O_2$ formation and OOH dissociation on $Pd_1/TiO_2$. **e** The reaction energy barriers of $H_2O_2$ formation and OOH dissociation on $Pd_8O_8/TiO_2$. **f** The reaction energy barriers of adsorbed $H_2O_2*$ dissociation steps on $Pd_1/TiO_2$ and $Pd_8O_8/TiO_2$. **g** The reaction energy barriers of OH* with H* on $Pd_1/TiO_2$ and $Pd_8O_8/TiO_2$. **h** The entire reaction potential energy landscape on $Pd_1/TiO_2$ and $Pd_8O_8/TiO_2$. **i** Schematic illustration of $H_2O_2$ formation on single Pd atom catalyst.

(Fig. 5f), suggesting that $H_2O_2$ is more difficult to degrade on $Pd_1/TiO_2$. Moreover, the energy barrier of the reaction of the OH* with H* on $Pd_1/TiO_2$ (0.83 eV) is higher than that on the $Pd_8O_8/TiO_2$ (0.50 eV) (Fig. 5g). Therefore, after the calculation of these transition states, the entire reaction potential energy landscape can be obtained (Fig. 5h). The results reveal that it is conducive to the generation of $H_2O_2$ on $Pd_1/TiO_2$ than on $Pd_8O_8/TiO_2$, which is well consistent with the experimental observations. Therefore, a schematic diagram of $H_2O_2$ formation on single Pd atom catalyst is shown in Fig. 5i.

To sum up, the DFT calculation results confirm that single Pd atom catalysts favor the formation of the key intermediate (*OOH) and $H_2O_2$, but strongly suppress the cleavage of O − O bond in $O_2$ and OOH and $H_2O_2$, leading to a higher activity and selectivity. The calculation results also show that the different performances come from various adsorption configurations of $O_2$ on single atom and clusters at the beginning.

In summary, we have synthesized a series of O-Pd/TiO₂ (oxidized) and M-Pd/TiO₂ (metallic) catalysts for the oxygen hydrogenation to $H_2O_2$, the catalytic performance of O-Pd/TiO₂ (including single Pd atom (0.1%), Pd cluster (1%)) are all better than that of M-Pd/TiO₂ catalyst. Particularly, the single Pd atom catalyst displays ultrahigh activity (115 mol/$g_{Pd}$/h), which is 14 and 135 times higher than that of clusters and nanoparticles, respectively; and the selectivity to $H_2O_2$ is more than 99%. More interesting, $H_2O_2$ degradation was also shut down. The concentration of $H_2O_2$ reached 1.07 wt.% in a batch. DFT

calculations show that the O−O bond breaking is significantly inhibited on the single Pd atom and the $O_2$ is easier to be activated to form *OOH and $H_2O_2$; and the energy barrier of $H_2O_2$ degradation is also higher. As a result, the high yield and selectivity is obtained on single Pd atom catalyst. The work reports the application of single Pd atom catalysts in the direct synthesis of $H_2O_2$. We believe it will yield far-reaching implications for subsequent catalyst design in direct synthesis of $H_2O_2$ and corresponding mechanism research in the future.

# Methods
## Materials
Titanium (IV) oxide, Aeroxide P25 (Beijing Balinwei Technology Co., Ltd, product of Japan), Ethylene glycol (A.R. Tianjin Damao Chemical Reagent Factory), $Na_2PdCl_4$ (>36.0%, Annege Chemical). Methyl alcohol (G.R. Tianjin Guangfu Science and Technology Development Co., Ltd). Fe $(NH_4)_2 \cdot (SO_4)_2 \cdot 6H_2O$ (Tianjin Institute of Guangfu Fine Chemicals). Cerium sulfate (macklin reagent). Ultrapure water (18.2 MΩ cm). Stainless steel autoclave (Yanzheng Shanghai Instrument Co., Ltd). 5% $H_2/CO_2$, 5% $H_2/N_2$, pure $N_2$ and 25% $O_2/CO_2$ were purchased from Beijing Millennium Capital Gas Co. Ltd.

## Synthesis of catalysts
Synthesis of O-Pd/TiO₂ (oxidized): TiO₂ was calcination at 450 °C for 4 h in the air (ramping rate: 5 °C/min) to remove surface water before catalysts preparation. The O-Pd/TiO₂ catalyst was

prepared using a hydrothermal synthesis method. The corresponding amount of $Na_2PdCl_4$ (10 g/L) was added to the $TiO_2$ (1 g) carrier suspension and heated to 80 °C for 3 h under magnetic stirring. Then, the obtained $Pd/TiO_2$ powder was centrifuged and freeze-dried overnight. Subsequently, the obtained catalyst was calcined at 350 °C for 3 h in the air at a heating rate of 5 °C/min. In order to obtain larger size palladium oxide nanoparticles, the samples with 3% palladium loading were heated at 450 °C and 600 °C for 3 h and other conditions remain unchanged. The prepared catalysts were denoted as 0.05%, 0.1%, 0.5%, 1%, 2% O-Pd/$TiO_2$ respectively according to the Pd loading. The catalysts with 3% Pd loading were labeled as 3% O-Pd/$TiO_2$−450 and 3% O-Pd/$TiO_2$−600, respectively, according to the calcination temperature.

Synthesis of M-Pd/$TiO_2$ (metallic): The preparation of M-Pd/$TiO_2$ was like that of O-Pd/$TiO_2$, except that the water was replaced with ethylene glycol and the temperature of $TiO_2$ carrier suspension was 100 °C, under reflux conditions. The catalysts were not calcined with air after centrifugally freeze-dried and used directly. The prepared catalysts were marked as 0.1,1,3, and 5% M-Pd/$TiO_2$ by the loading of palladium.

## Characterization

The Pd loading on the catalysts was analyzed using an IRIS Intrepid ER/S (Thermo Elemental) inductively coupled plasma-atomic emission spectrometer (ICP-AES). Transmission electron microscopy (TEM) images were obtained using a Tecnai F20 microscope in conjunction with powder samples deposited onto a copper micro-grid and coated with carbon, applying an accelerating voltage of 200 kV. Spherical aberration corrected (Cs corrected) high angle annular dark field scanning transmission electron microscopy (HAADF-STEM) and energy-dispersive X-ray (EDX) mapping images were obtained with an FEI Titan G2 microscope equipped with a Super-X detector, operating at 300 kV. X-ray diffraction (XRD) patterns of the samples were recorded on a Bruker D8 Advance system with Cu Kα radiation at 40 kV and 40 mA. X-ray photoelectron spectroscopy (XPS) analyses were performed using a Kratos Axis Ultra XPS spectrometer with monochromatized Al-Kα radiation and an energy resolution of 0.48 eV. The X-ray absorption fine structure (XAFS) spectrum data were collected on the BL14W1 beamline radiation equipment of the Shanghai Synchrotron Radiation Facility (SSRF) of the Shanghai Institute of Applied Physics (SINAP). Pd foil, PdO samples were used as references. All target samples and references were measured by fluorescence or transmission mode. Extended X-ray adsorption fine structure (EXAFS) fitting was conducted using the software of Artemis.

## Catalytic experimental measurement

Catalytic experiments were operated using a stainless-steel autoclave with a nominal volume of 50 mL and a maximum working pressure of 14 MPa.

$H_2O_2$ synthesis. In the typical experiment of $H_2O_2$ synthesis, 2.5 mg catalyst and 8.5 g solvent (5.6 g $CH_3OH$ (HPLC grade), 2.9 g $H_2O$) were added into the autoclave. Before 3.0 MPa 5%$H_2$/$CO_2$ was injected at room temperature, the reactor was purged three times with 0.7 MPa (5%$H_2$/$CO_2$), then the temperature was reduced to 2 °C (in an ice bath), the pressure was about 2.3 MPa, at this time, 1.2 MPa 25% $O_2$/$CO_2$ was injected, and the total pressure was 3.5 MPa. Temperature and pressure are respectively detected by thermocouple and pressure sensor. The stirring speed was controlled at 1200 rpm.

The $H_2O_2$ yield was detected by acidified Ce ($SO_4$)$_2$ (0.01 M) titration in the presence of two drops of ferroin indicator. The Ce ($SO_4$)$_2$ solutions were standardized against $(NH_4)_2Fe(SO_4)_2·6H_2O$ using ferroin as an indicator. Catalyst yields are marked as mol $H_2O_2$ $kg_{cat}^{-1}h^{-1}$

according to the following equation.

$$H_2O_2 \text{ Yield} = (\text{moles of } H_2O_2 \text{ generated})/(\text{mass of catalyst} \times \text{time}) \quad (1)$$

Gas analysis was performed by gas chromatography (GC-2020) equipped with a TDX-01 column connected to a thermal conductivity detector. Conversion of $H_2$ was calculated by gas analysis before and after the reaction. $H_2O_2$ selectivity was calculated according to the following equation:

$$H_2O_2 \text{ Selectivity} = (\text{moles of } H_2O_2 \text{ generated})/(\text{mass of } H_2 \text{ reacted}) \times 100\%$$
$$(2)$$

$H_2O_2$ degradation. The experiments were manipulated in a similar way to the $H_2O_2$ synthesis, but in the absence of 1.2 MPa 25%$O_2$/$CO_2$. In detail, $H_2O$ from the 8.5 g of solvent was replaced by a 30% $H_2O_2$ solution to give a reaction solvent containing between 2-8 wt%$H_2O_2$. The standard reaction conditions adopted for $H_2O_2$ degradation were as follows: 2.5 mg catalyst, 8.5 g solvent (5.6 g $CH_3OH$, 2.34 g $H_2O$, and 0.56 g 30% $H_2O_2$, 3.0 MPa 5%$H_2$/$CO_2$, 2 °C, 1200 rpm, 30 min). The $H_2O_2$ degradation rate is a combination of $H_2O_2$ hydrogenation and decomposition. $H_2O_2$ degradation rate was calculated following:

$$H_2O_2 \text{ degradation rate} = (\text{moles of } H_2O_2 \text{ reduced})/(\text{mass of catalyst} \times \text{time})$$
$$(3)$$

To ensure the reliability of the data, all the above experiments have to be tested for nine times, the data presented was the average value, the error of $H_2O_2$ yield and selectivity are within 1%, 4% respectively.

## Calculation details

All the spin-polarized density functional (DFT) calculations were carried out with the Vienna Ab initio Simulation Package(VASP)[39]. The ion-electron interaction was described by the projector augmented wave (PAW) method[40]. The generalized gradient approximation (GGA) in the Perdew-Burke-Ernzerhof (PBE) functional was used for the exchange-correlation interactions[41]. The DFT-D3 method was introduced to describe van der Waals interactions[42]. The 15 Å vacuum slab in the z direction was applied to avoid interactions with adjacent units. The cut-off energy for plane-wave basis set was 450 eV. The convergence criterion for energy and force were set as 10-5 eV and 0.03 eV/Å during geometry optimization, respectively. The Brillouin zone was sampled with $2 \times 2 \times 1$ k-point grids. Transition states were located with the climbing-image nudged elastic band (CI-NEB) method[43], and the threshold value for the forces on each atom was 0.05 eV/Å.

## Data availability

The data supporting this study are available within the paper and the Supplementary Information. All other relevant source data are available from the corresponding authors upon reasonable request. Source data are provided with this paper.

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

## Acknowledgements

This work was supported by the National Natural Science Foundation of China (NSFC12075015 to G.C.) and Beijing Municipal Great Wall Scholar Training Plan Project (CIT&TCD20190307 to H.L.). The authors thank the BL14W1 beamline radiation equipment of Shanghai Synchrotron Radiation Facility (SSRF) of Shanghai Institute of Applied Physics (SINAP).

## Author contributions

S.Y. Conceptualization, Methodology, Investigation, Writing-original draft. X.C. Validation, Formal analysis, Data curation. Y.W. Validation,

Data curation. B.X. Data curation. Y.X. Validation, Formal analysis. J.R. DFT calculation. Y.L. Data curation. H.L. Resources, Data curation. C.Z. Writing—review and editing. G.C. Writing—review and editing, Supervision.

## Competing interests

The authors declare no competing interests.

## Additional information

**Supplementary information** The online version contains

supplementary material available at

