## [Peer Review File · Nature Communications]

Title: High activity and selectivity of single palladium atom for oxygen hydrogenation to H₂O₂REVIEWER COMMENTS

Reviewer #1 (Remarks to the Author):

This manuscript reports on the use of various Pd catalysts for the direct synthesis of H₂O₂ from H₂ and O₂. The authors find that oxidized Pd species are far more selective for H₂O₂ synthesis, and that atomically dispersed O-Pd species are particularly active. The reports of very high selectivity and good activity on a Pd metal basis are very interesting and can potentially have a significant impact on the field, but there are some aspects of the report that should be improved prior to publication.

(1) The treatment of the kinetics is somewhat qualitative, making it confusing to try to understand how all the pathways are connected. The authors do report yields (or reaction rates) in useful units, but the origins of the effects of reaction time and catalyst loading are not very clear. One interpretation of the effects of both time and loading is that as conversion increases, selectivity decreases due to H₂O₂ consumption. This is the implicit idea behind the experiment described in Figure 3e, but it is not very clear whether the observed rates of H₂O₂ consumption are consistent with the observed effects of catalyst mass and reactor time, since the results are presented in terms of moles consumed. Put another way: In scheme 1, the authors show that competitive rates for four different reaction processes would be responsible for trends in yield and selectivity, but their analysis in Figure 3 does not make it clear what effect each catalyst composition has on the relative rates of the corresponding processes. I think there are many ways the authors could address this, and they likely already have the data -- reporting selectivity as a function of conversion for the different catalysts could help.

(2) The authors reported on an experiment in which they "refreshed" the gas used in the reactor to increase yield. Would higher H₂ partial pressures also lead to higher yields?

(3) In Table 1, the authors report that the selectivity error (based on 9 replicates) was 4%. Can they also report the yield errors? Can they report fractional conversions for H₂ and/or O₂ in the same table?

(4) It's not very clear that the model used in the DFT calculations is representative of the actual catalyst surface. For example, it a barrier of 0.81 eV for endothermic OOH formation consistent with a high observed rate for H₂O₂ formation in the experiments. This seems like a quite high barrier. Could the authors make a more quantitative comparison to experiments?

(5) Are the active site models for the Pd single-atom catalyst shown in the SI stable compared to other possible configurations?

(6) It would help if H₂O₂ desorption were included on the plots on Figure 4.

(7) The Discussion section does not add very much and could be eliminated from the manuscript entirely.

(8) In one place in the manuscript, the authors refer to hydrogen activation as a key step that is easier on a reduced Pd site. I'm not sure I understand how they are proposing that H₂ gets activated, since it seems to already exist as H* for the DFT calculations. Could the authors perhaps show a scheme of a catalytic cycle for what they are proposing as far as how the single atom catalyst functions?

Reviewer #2 (Remarks to the Author):

This manuscript synthesized oxidized single-atom Pd/TiO₂ catalyst for oxygen reduction to H₂O₂. Using only Pd catalyst, the authors developed high performance catalyst and they characterized the catalyst using diverse methods such as XAS and DFT. Numeric data of catalytic activity seems attractive, however it is difficult to recommend this version of study to Nature Communications. Considering the quality required to be published in Nature Communications, data quality is insufficient and outperformed catalytic activity, major strong point of this study, is not enough to be superior to other studies in DSHP field. Detail comments are listed here below.

Comments:

1. In title, "High" could be better than "Highly".
2. Figure quality is insufficient to clearly deliver the authors' claims.
3. Different amount of Pd made gradual TiO₂ peak shift. Since bulk phase showed this distinguishable shifts, Pd might be located not only the surface, and this is also important factor. I recommend the authors to investigate this situation.
4. Due to low resolution, single atom Pd cannot be observed in Figure 1b. They should be recognizable even without the red circles.
5. In EXAFS data, detail explanation for how the coordination number of 4 is assigned to atomically dispersed Pd on TiO₂ is required.
6. In DFT calculation, electronic states of Pd single atom should be checked. It couldn't be close to 2+ (PdO state) rather to 1+.
7. Due to low resolution, I cannot distinguish each spectrum in Figure 2a.
8. Recent studies reveal that Pd can easily be reduced under DSHP reaction conditions, which is in well agreement in the authors Fig S5 (they reduced during the reaction). Under mild condition with low H₂ ratio and pressure, perfectly oxidized Pd-O catalyst cannot show H₂ conversion, indicating that the authors not analyzed "real-states in catalysis reaction" but intensively analyzed fresh states of O-Pd/TiO₂.
9. In terms of superior catalytic performance. The authors performed their tests under almost same reaction conditions to Hutchings group [26 FEBRUARY 2016 • VOL 351 ISSUE 6276]. So I can calculate H₂ conversion via given data. Hutchings group's 3 wt% Pd-2 wt% Sn/TiO₂ catalyst showed 61 mol/kgcat/h with 96% of H₂O₂ selectivity using 10 mg of catalyst. And they gave H₂ conversion 9%. The authors used 2.5 mg of 0.1 wt% O-Pd/TiO₂ catalyst and achieved 115 mol/kgcat/h with 99% of selectivity. It could be suggested that O-Pd/TiO₂ herein achieved 2.25 % of H₂ conversion which is extremely insufficient to

demonstrate powerful catalytic activity even in high-pressure batch reactor. In other words, formidably small amount of catalyst can achieve both significantly high H₂O₂ production, high H₂O₂ selectivity, and low H₂O₂ decomposition. I agree with the catalytic activity is attractive and excellent, however it would not be recommended to high impact journal like Nature Communications.

10. In similar thought, H₂O₂ degradation can occur by metal oxides, especially by TiO₂ [Decomposition of Hydrogen Peroxide at Water-Ceramic Oxide Interfaces]. The authors claimed 0.1 wt% O-Pd/TiO₂ shut down the degradation, however it also could be derived by low amount of catalyst. Since 0.1 wt% is insufficient to cover the surface as described in Fig 2b TEM image, a number of TiO₂ sites can be exposed during the reaction. We can infer that this did not happen under this reaction condition.

Author's Response to Reviewers

Reviewer 1

This manuscript reports on the use of various Pd catalysts for the direct synthesis of H₂O₂ from H₂ and O₂. The authors find that oxidized Pd species are far more selective for H₂O₂ synthesis, and that atomically dispersed O-Pd species are particularly active. The reports of very high selectivity and good activity on a Pd metal basis are very interesting and can potentially have a significant impact on the field, but there are some aspects of the report that should be improved prior to publication.

Response: We thank the reviewer's constructive comments and recognition of our work. We would like to address those comments as below.

Comment 1. The treatment of the kinetics is somewhat qualitative, making it confusing to try to understand how all the pathways are connected. The authors do report yields (or reaction rates) in useful units, but the origins of the effects of reaction time and catalyst loading are not very clear. One interpretation of the effects of both time and loading is that as conversion increases, selectivity decreases due to H₂O₂ consumption. This is the implicit idea behind the experiment described in Figure 3e, but it is not very clear whether the observed rates of H₂O₂ consumption are consistent with the observed effects of catalyst mass and reactor time, since the results are presented in terms of moles consumed. Put another way: In scheme 1, the authors show that competitive rates for four different reaction processes would be responsible for trends in yield and selectivity, but their analysis in Figure 3 does not make it clear what effect each catalyst composition has on the relative rates of the corresponding processes. I think there are many ways the authors could address this, and they likely already have the data -- reporting selectivity as a function of conversion for the different catalysts could help.

Response: Thanks for the constructive suggestions. As suggested by the reviewer, with the increase of H₂ conversion, the selectivity decreases due to H₂O₂ degradation. This could be better understood by comparing selectivity as a function of conversion for the different catalysts (Supplementary Fig. 5). Thank you again for your valuable comments.

Supplementary Fig. 5. H₂ conversion and H₂O₂ selectivity with different reaction time on various loading catalysts (catalyst feeding: 10 mg).

Comment 2. The authors reported on an experiment in which they “refreshed” the gas used in the reactor to increase yield. Would higher H₂ partial pressures also lead to higher yields?

Response: Thanks for the constructive suggestions. We used 3.0 Mpa 5% H₂/CO₂, 1.2 Mpa 25% O₂/CO₂ gas for the first ventilation, and the pressure decrease to 2.7 Mpa after 2.5 hours reaction due to gas consumption, which might affect the further H₂O₂ yield. As suggested by the reviewers, higher H₂ partial pressure may lead to higher yield. So we refresh the same gas to renew the H₂ partial pressures, and thus to improve the final H₂O₂ yield. Thank you again for your valuable comments.

Comment 3. In Table 1, the authors report that the selectivity error (based on 9 replicates) was 4%. Can they also report the yield errors? Can they report fractional conversions for H₂ and/or O₂ in the same table?

Response: Thanks for the constructive suggestions. The error of H₂O₂ yield and selectivity are within 1% and 4% respectively. H₂ conversion has been added to Table 1 in the revised manuscript.

Comment 4. It’s not very clear that the model used in the DFT calculations is representative of the actual catalyst surface. For example, it a barrier of 0.81 eV for endothermic OOH formation consistent with a high observed rate for H₂O₂ formation in the experiments. This seems like a quite high barrier. Could the authors make a more quantitative comparison to experiments?

Response: Thanks for the constructive suggestions. We have carefully checked the calculated energy barrier, and the lowest barrier is still 0.81 eV. On the other hand, the energy barrier of 0.81eV is close to other reported values; for example, the OOH formation barrier on Pd (111) and (100) is 0.92 eV and 1.11 eV, respectively (J. Phys. Chem. C 2019, 123, 26324–26337). In this regard, a barrier of 0.81 eV might be reasonable. Thank you again for your valuable comments.

Comment 5. Are the active site models for the Pd single-atom catalyst shown in the SI stable compared to other possible configurations?

Response: Thanks for the constructive suggestions. The EXAFS fitting results showed the coordination number of Pd-O is 4 for 0.1%O-Pd/TiO₂ (Supplementary Table1), representative of the Pd single atoms on the surface of TiO₂ (Science, 2016, 352(6287):797-801; Chinese Journal of Catalysis. 2017,38, 1574-1580). So we think the model might be a stable configuration. Thank you again for your valuable comments.

Comment 6. It would help if H₂O₂ desorption were included on the plots on Figure 4.

Response: Thanks for the constructive suggestions. H₂O₂ desorption were included on the plots on Fig. 4f. Thank you again for your valuable comments.

Figure 4. f) The reaction energy barriers of adsorbed H₂O₂* dissociation steps on Pd₁/TiO₂ and Pd₈O₈/TiO₂.

Comment 7. The Discussion section does not add very much and could be eliminated from the manuscript entirely.

Response: Thanks for the constructive suggestions. The discussion section has been deleted from our revised manuscript.

Comment 8. In one place in the manuscript, the authors refer to hydrogen activation as a key step that is easier on a reduced Pd site. I'm not sure I understand how they are proposing that H₂ gets activated, since it seems to already exist as H* for the DFT calculations. Could the authors perhaps show a scheme of a catalytic cycle for what they are proposing as far as how the single atom catalyst functions?

Response: Thanks for the constructive suggestions. We are sorry for our unclear expression in the original manuscript, what we trying to say is that although the reduced Pd sites activate H₂ easily, it also smoothly dissociate the O-O bond. As to the H₂ activation on the catalyst, we have supplemented the calculation of H₂ dissociation energy of Pd₁/TiO₂ and Pd₈O₈/TiO₂ (Fig. 4b, Fig. 4h) in the revised manuscript. The results show that H₂ is more easily activated on Pd₁/TiO₂ than Pd₈O₈/TiO₂. A recent work also suggest that oxidized Pd species can dissociate H₂ well (J. Am. Chem. Soc. 2019, 141, 901-910). Relative description and discussion are supplemented in the main text in the revised manuscript; besides, a schematic diagram of H₂O₂ formation on single Pd atom is shown in Fig. 4i. Thank you again for your valuable comments.

Figure 4. b) Energy profiles for H₂ dissociation on Pd₁/TiO₂ and Pd₈O₈/TiO₂. h) The entire reaction potential energy landscape on Pd₁/TiO₂ and Pd₈O₈/TiO₂. i) Schematic illustration of H₂O₂ formation on single Pd atom catalyst.

Reviewer 2

This manuscript synthesized oxidized single-atom Pd/TiO₂ catalyst for oxygen reduction to H₂O₂. Using only Pd catalyst, the authors developed high performance catalyst and they characterized the catalyst using diverse methods such as XAS and DFT. Numeric data of catalytic activity seems attractive, however it is difficult to recommend this version of study to Nature Communications. Considering the quality required to be published in Nature Communications, data quality is insufficient and outperformed catalytic activity, major strong point of this study, is not enough to be superior to other studies in DSHP field.

Response: We thank the reviewer's constructive comments and important concerns on our manuscript. We would like to address those comments as below.

Comment 1. In title, "High" could be better than "Highly".

Response: Thanks for the suggestions. "Highly" in the title has been replaced by "High" in our revised manuscript.

Comment 2. Figure quality is insufficient to clearly deliver the authors' claims.

Response: Thanks for the suggestions. We are not sure whether this comment is concerned with our figure resolution or the data quality. If the former, we have updated the picture resolution in the revised manuscript. If the latter, we will respond in detail to the following comments.

Comment 3. Different amount of Pd made gradual TiO₂ peak shift. Since bulk phase showed this distinguishable shifts, Pd might be located not only the surface, and this is also important factor. I recommend the authors to investigate this situation.

Response: Thanks for the constructive suggestions. As mentioned by the reviewer, the peak of TiO₂ may shift when palladium enters the bulk phase. However, the diffraction peaks of TiO₂ of different samples have showed no obvious shift compared with pure TiO₂ in the XRD patterns (Fig.1a). Since the material synthesis in our experiment is a simple and mild wet

impregnation, most Pd species might be on the surface of TiO₂, which agreed well with other reported mild synthesized conditions (ACS Catal. 2020, 10, 12932–12942; J. Am. Chem. Soc. 2018, 140, 954–962; Science, 2016, 352, 797-801; Chinese Journal of Catalysis. 2017, 38, 1574-1580). The author think the problem is very interesting, but it is not within the scope of this work. Thank you again for your valuable comments.

Comment 4. Due to low resolution, single atom Pd cannot be observed in Figure 1b. They should be recognizable even without the red circles.

Response: Thanks for the suggestions. We have updated the resolution of the figure in the revised manuscript. Because of the low z-contrast between Pd ($z = 46$) and Ti ($z = 22$), the single Pd atom in Figure 1b may not be very clear on the TiO₂ support. In the Pd₁/CeO₂ system, single Pd atoms could not even be seen under AC-TEM (Angew. Chem. Int. Ed. 2021, 60, 22769–22775). However, we also confirm the existence of isolated Pd atom by using EXAFS fitting results which is discussed in detail in the response to the comment 5.

Comment 5. In EXAFS data, detail explanation for how the coordination number of 4 is assigned to atomically dispersed Pd on TiO₂ is required.

Response: Thanks for the suggestions. We are sorry for it may not be clearly expressed in the original manuscript. According to our EXAFS fitting results (Fig. 2c, Supplementary Table1), the coordination number of Pd-O is 4 for 0.1% O-Pd/TiO₂, and there is no Pd-Pd bond for 0.1% O-Pd/TiO₂ (Fig. 2b, Supplementary Table1), representative of the Pd single atom on the surface of TiO₂ (Science, 2016, 352(6287):797-801; Chinese Journal of Catalysis. 2017,38, 1574-1580).

Comment 6. In DFT calculation, electronic states of Pd single atom should be checked. It couldn' t be close to 2+ (PdO state) rather to 1+.

Response: Thanks for the suggestions. The oxidation state of Pd single atom estimated by bader charge analysis is (+1.73) between +1 and +2 (Supplementary Fig. 10), which agreed well with XAS results (Fig. 2a).

Comment 7. Due to low resolution, I cannot distinguish each spectrum in Figure 2a.

Response: Thanks for the constructive suggestions. We have updated the resolution of the figure in the revised manuscript.

Figure 2 a.) XANES spectra at the Pd K-edge

Comment 8. Recent studies reveal that Pd can easily be reduced under DSHP reaction conditions, which is in well agreement in the authors Fig S5 (they reduced during the reaction). Under mild condition with low H₂ ratio and pressure, perfectly oxidized Pd-O catalyst cannot show H₂ conversion, indicating that the authors not analyzed “real-states in catalysis reaction” but intensively analyzed fresh states of O-Pd/TiO₂.

Response: Thanks for the constructive suggestions. The concerns mentioned by the reviewer are very forward-looking. As to the H₂ activation on oxidized Pd-O catalyst, we have supplemented the calculation of H₂ dissociation energy on Pd₁/TiO₂ and Pd₈O₈/TiO₂ (Fig. 4b, Fig. 4h) in the revised manuscript. The results show that H₂ is more easily activated on Pd₁/TiO₂. A recent work also reported that oxidized Pd can also dissociate H₂ well (J. Am. Chem. Soc. 2019, 141, 901-910). As to “real-states”, the investigation of “real-states in catalysis reaction” is very important for novel catalyst design as well as the understanding of reaction mechanism. The operando spectroscopy/TEM techniques are often needed to unveil the “real states” of catalyst. However, the reaction condition are harsh for direct synthesis of H₂O₂ (3.5 MPa), make it difficult to handle the operando techniques in the experiment.

Nevertheless, the work is the first report on the direct synthesis of H₂O₂ by single Pd atom catalyst which exhibited high activity (115 mol/g_{Pd}/h) and selectivity(>99%). In our humble opinion, it is a relatively systematic and intact work, and we are very willing to be committed to study the real-states of catalyst in the direct synthesis of H₂O₂ with the development of operando techniques in the future. Thank you again for your valuable comments.

Comment 9. In terms of superior catalytic performance. The authors performed their tests under almost same reaction conditions to Hutchings group [26 FEBRUARY 2016 • VOL 351 ISSUE 6276]. So I can calculate H₂ conversion via given data. Hutchings group's 3 wt% Pd-2 wt% Sn/TiO₂ catalyst showed 61 mol/kg_{cat}/h with 96% of H₂O₂ selectivity using 10 mg of catalyst. And they gave H₂ conversion 9%. The authors used 2.5 mg of 0.1 wt% O-Pd/TiO₂ catalyst and achieved 115 mol/kg_{cat}/h with 99% of selectivity. It could be suggested that O-Pd/TiO₂ herein achieved 2.25 % of H₂ conversion which is extremely insufficient to demonstrate powerful catalytic activity even in high-pressure batch reactor. In other words, formidably small amount of catalyst can achieve both significantly high H₂O₂ production, high H₂O₂ selectivity, and low H₂O₂ decomposition. I agree with the catalytic activity is attractive and excellent, however it would not be recommended to high impact journal like Nature Communications.

Response: Thanks for the comment. The reviewer carefully compared our data with Hutchings' data and concluded that the H₂ conversion rate was low in our work. In our humble opinion, it may not be most appropriate to compare H₂ conversion between a small number of catalysts (2.5mg in our experiment) and a large number of catalysts (10mg in the Hutchings' work); actually, we have investigated the effect of catalyst feeding (2.5, 5, 10 mg) on the catalytic performance in the original manuscript. (Fig.3a, 3b); for 10 mg 0.1% O-Pd/TiO₂ catalyst feeding, the calculated H₂ conversion is 10.43% which is compared to the reviewer mentioned PdSn catalyst. (9% of H₂ conversion using 10 mg of 3 wt% Pd-2 wt% Sn/TiO₂). Besides, the H₂O₂ yield and selectivity may be more commonly used to evaluate catalyst than the H₂ conversion.

Comment 10. In similar thought, H₂O₂ degradation can occur by metal oxides, especially by TiO₂ [Decomposition of Hydrogen Peroxide at Water-Ceramic Oxide Interfaces]. The authors claimed 0.1 wt% O-Pd/TiO₂ shut down the degradation, however it also could be derived by low amount of catalyst. Since 0.1 wt% is insufficient to cover the surface as described in Fig 2b TEM image, a number of TiO₂ sites can be exposed during the reaction. We can infer that this did not happen under this reaction condition.

Response: Thanks for the comment. We thank the reviewer recommending us an important literature [Decomposition of Hydrogen Peroxide at Water-Ceramic Oxide Interfaces. J. Phys. Chem. B 2005, 109, 3364-3370] and we have carefully read the literature. We have noticed the H₂O₂ decomposition tests were performed in the temperature range of 25 °C to 120 °C, which is higher than our reaction temperature of 2 °C. In our experiment, the H₂O₂ degradation rate was not detected on pure TiO₂ (P25) sample at 2 °C (Supplementary Table 2) which is consistent with the results of other work. (Science 2009, 323, 1037-1041, Table 1 showed in the following). In compared with pure TiO₂ and O-Pd/TiO₂, the H₂O₂ degradation rate can be obviously detected on the M-Pd/TiO₂ catalyst (Table1, Entry 9-11) in our experiment, note the catalyst feeding is still “2.5 mg”; thus, the observed H₂O₂ degradation shut down in our experiment might not be due to the low amount of catalyst (2.5 mg). Furthermore, the H₂O₂ degradation rate was still not detected by using up to “30 mg” 0.1 wt% O-Pd/TiO₂ catalyst. Combined with other H₂O₂ degradation results in our experiment (Supplementary Table 4), in our humble opinion, the H₂O₂ degradation hardly occurs on the O-Pd/TiO₂ catalyst. Thank you again for your valuable comments.

Supplementary Table 2 H₂O₂ degradation test on TiO₂ support itself ^a

TiO ₂	H ₂ O ₂ degradation (mol/kg _{cat} /h)						
	0.5 h	1.0 h	1.5 h	2.0 h	2.5 h	3.0 h	
	n.d	n.d	n.d	n.d	n.d	n.d	

a: H₂O₂ degradation was under standard reaction conditions: 5% H₂/CO₂ (3.0 MPa), 8.5 g solvent (2% H₂O₂-8% H₂O), 2.5-10 mg TiO₂, 2°C, 1200 rpm, reaction time: 0.5 hour to 3 hours. n.d=not detected. 2% H₂O₂(5.6 g CH₃OH,2.34 g H₂O, and 0.56 g 30% H₂O₂).4% H₂O₂(5.6g CH₃OH,1.77 g H₂O, and 1.13 g 30% H₂O₂);

8% H₂O₂(5.6g CH₃OH,0.63 g H₂O, and 2.27 g 30% H₂O₂)

Table 1. Effect of acid pretreatment of the support on the hydrogenation and decomposition of H₂O₂.

Support	H ₂ O ₂ degradation		Support only	
	Untreated		Pretreated†	
	Hydrog‡	Decomp§	Hydrog‡	Decomp§
Al ₂ O ₃	0	0	0	0
TiO ₂	0	0	0	0
SiO ₂	0	0	0	0
Carbon	4	1	3	1

The Table 1 above comes from Science. 2009, 323, 1037-1041

REVIEWER COMMENTS

Reviewer #1 (Remarks to the Author):

The changes made by the authors are adequate to justify publication. The treatment of kinetics remains essentially qualitative, as does the connection with the DFT calculations, but the main results are important and the work seems to be reported in such a way that it can readily be reproduced or further analyzed by others.

Reviewer #2 (Remarks to the Author):

I appreciate to the authors that they demonstrated the novelty of their study and efforts to make reasonable responses to my comments. Resolution of the Figures is improved and H₂ conversion data with different amount of catalyst use impressively appeal the high performance of the catalysts. I also understand the difficulties for some explanation, however as the journal is Nature Communications, some comments and investigation seem further to be necessary.

Comment 1. In the manuscript, there are some typos to be corrected such as “experiements” and spacing of the words in Figure 3 caption.

Comment 2. In Figure 1b and Figure S7, I recommend the authors to delete the red circles. Although I understand the authors’ explanation (hard to clearly confirm the single Pd atoms), but then I rather recommend not to suggest unobservable sites by red circles.

Comment 3. Second one is “clear correlation between the catalyst and characterizations”. In my humble opinion, “the active site of 0.1%O-Pd/TiO₂ could be changing during the reaction”. Related comments are as follows:

- 1) During 2.5 hours reaction, the H₂O₂ selectivity of 0.1%O-Pd/TiO₂ is almost maintained (Figure 3d), but the H₂O₂ yield gradually decreased as it recycled (Figure 3f). Would the authors explain the reason for this situation? In the recycle tests are the same amounts of catalysts used?
- 2) After the reaction, as Figure S6 shows, palladium electronic states obviously shift to metallic phase compared to the as-prepared oxidized state in Figure 2d, indicating that the palladium state was changing during the reaction. Nevertheless, the catalyst still maintain ~100% H₂O₂ selectivity during the 2.5 hours reaction. Then, could we demonstrate the 100% of H₂O₂ selectivity dominantly resulted from the intrinsic property of the catalyst? Catalyst was changed but the activity was maintained, which seems less reasonable.
- 3) After the reaction, how does the single Pd show metallic phase despite undetectable single Pd sites in Figure S7?
- 4) In DFT calculations, 0.58 eV of H₂O₂ decomposition barrier can also be progressed since a higher 0.81 eV of OOH formation on single Pd was also progressed. However, in the authors’ test results, the catalyst with single Pd didn’t decompose H₂O₂. Please explain this inconsistency between DFT and

experimental results.

5) The reviewer understands the difficulties of operando analysis. Although Flaherty group did the high pressurized operando analysis, the reviewer do not strongly recommend it this study. However, the key is which state dominated the catalytic activity, before the reaction (oxidized Pd) or after the reaction (metallic Pd)? It should be examined by reasonable evidences.

Comment 4. In DFT study section, it would be better to change "cluster" to "PdO cluster".

Comment 5. The authors responded that XRD peak shift is not obvious, however only the 0.1%O-Pd/TiO₂ shows the shift as below. Since others are perfectly matched, it would provide information that 0.1%O-Pd/TiO₂ might have different lattice structure compared to others. Please reconsider this

Author's Response to Reviewers

Reviewer 1

The changes made by the authors are adequate to justify publication. The treatment of kinetics remains essentially qualitative, as does the connection with the DFT calculations, but the main results are important and the work seems to be reported in such a way that it can readily be reproduced or further analyzed by others.

Response: We want to thank the reviewer for the thoughtful review and comments on our work.

Reviewer 2

I appreciate to the authors that they demonstrated the novelty of their study and efforts to make reasonable responses to my comments. Resolution of the Figures is improved and H₂ conversion data with different amount of catalyst use impressively appeal the high performance of the catalysts. I also understand the difficulties for some explanation, however as the journal is Nature Communications, some comments and investigation seem further to be necessary.

Response: Once again, we thank the reviewer's constructive comments and reasonable concerns on our manuscript which are very helpful for revising and improving our manuscript. We would like to address those comments as below.

Comment 1. In the manuscript, there are some typos to be corrected such as "experiements" and spacing of the words in Figure 3 caption.

Response: Thanks for the comment. We have carefully checked the revised manuscript, and the typos are corrected.

Comment 2. In Figure 1b and Figure S7, I recommend the authors to delete the red circles. Although I understand the authors' explanation (hard to clearly confirm the single Pd atoms), but then I rather recommend not to suggest unobservable sites by red circles.

Response: Thanks for the comment. We have deleted the red circles in the Figure 1b and

Supplementary Fig. 6.

Comment 3. Second one is “clear correlation between the catalyst and characterizations”. In my humble opinion, “the active site of 0.1%O-Pd/TiO₂ could be changing during the reaction”.

Related comments are as follows:

1) During 2.5 hours reaction, the H₂O₂ selectivity of 0.1%O-Pd/TiO₂ is almost maintained (Figure 3d), but the H₂O₂ yield gradually decreased as it recycled (Figure 3f). Would the authors explain the reason for this situation? In the recycle tests are the same amounts of catalysts used?

Response: Thanks for the constructive suggestions. We believe that the high H₂O₂ selectivity is due to the geometric structure of single Pd atom which is supported by experiments and DFT calculations in the manuscript. After reaction, we have not observed an obvious single Pd atom agglomeration under aberration-corrected transmission electron microscopy (Supplementary Fig. 6), indicating the geometric structure of single Pd atom remain unchanged. Thus, the selectivity does not change.

The operation of recycle experiment is as follows: in the recycle reaction, the amount of catalyst is constant (2.5mg); after the first reaction round, the catalyst is collected by centrifugation, washed with pure water three times, and then recycled directly. Since the catalyst loading mass is very little (2.5mg), it's reasonable to assume a delicate mass loss during the centrifugation and washing steps will affect the final H₂O₂ yield; thus, we believe the observed H₂O₂ yield decrease might be partially due to the catalyst loading loss in the recycle test. Another possible reason might be the very limited single Pd atom agglomeration occurred in the reaction, although we haven't observed an obvious agglomeration under aberration-corrected transmission electron microscopy (Supplementary Fig. 6).

2) After the reaction, as Figure S6 shows, palladium electronic states obviously shift to metallic phase compared to the as-prepared oxidized state in Figure 2d, indicating that the palladium state was changing during the reaction. Nevertheless, the catalyst still maintain ~100% H₂O₂ selectivity during the 2.5 hours reaction. Then, could we demonstrate the 100% of H₂O₂ selectivity dominantly resulted from the intrinsic property of the catalyst? Catalyst was changed but the activity was maintained, which seems less reasonable.

Response: Thanks for the constructive suggestions. As we have responded to the comment 3.1, the H₂O₂ selectivity is originated from the geometric structure of single Pd atom, which inhibits side reactions and H₂O₂ degradation. And the geometric structure of single Pd atom remains unchanged during the reaction, so the selectivity does not change. However, the change of Pd electronic structure (valence state) is due to the reductive condition in the reaction, and the as-prepared oxidized state of Pd single atom is reduced to a "near zero valence" state of single Pd atom in the *in-situ* catalytic reaction, and act as "active site" for the direct synthesis of H₂O₂. To verify this viewpoint, we have supplemented the XPS of 0.1%O-Pd/TiO₂ sample after the first round as well as that after 2.5 hours. The results show that the valence state of Pd species changes from oxidation state to "near zero valence" state after the first round (Supplementary Fig. 8), which suggests the "near zero valence" state of single Pd atom is active site. It is worth noting that the "near zero valence" state of single Pd atom derived in the *in-situ* catalytic reaction is quite different from the metallic-palladium (M-Pd) catalyst synthesized by a solvothermal method using reductive solvent (ethylene glycol) which is also reported in the manuscript. For example, both the H₂O₂ yield and selectivity of 0.1% and 1%M-Pd/TiO₂ are very poor (Table 1), which suggests the initial oxidation state of Pd atom is vital to catalytic performance. The result also agrees well with other work. (Journal of Catalysis 2012, 292, 227; ACS Nano 2012, 6, 6600; ACS Catalysis 2018, 8, 3418; National Science Review 2018, 5, 895.). Related description and discussion are supplemented in the revised manuscript.

Supplementary Fig. 8. Pd 3d XPS spectra of 0.1%O-Pd/TiO₂ catalysts (fresh sample, after 30 mins and 2.5 h reactions).

3) After the reaction, how does the single Pd show metallic phase despite undetectable single Pd sites in Figure S7?

Response: Thanks for the comment. This is a very good question. Although most of noble metal (Pt, Pd, etc.) single atom/cluster on the non-carbon support (oxide etc.) display oxidation state due to the high electronegativity of ligand atom, recent research suggests that the “near zero valence” state of noble metal single atom could be realized by manipulation of metal-support interactions (MSIs) [1-6]. Particularly, the oxygen vacancies in the support play an important role; Wei et al report the oxygen vacancies in the titanium oxide induce the charge transfer from support to Pt nanoclusters [5], and Cheng et al reveal that the oxygen vacancies in titanium oxide led to the lattice distortion, and thus affect the electronic structure of supported Pt single atom [6]. In our work, it is reasonable to assume that the hydrogen spillover occurred in the reaction might result in the oxygen vacancies in the TiO₂, and thus, affect the electronic structure of the single Pd atom. As a result, the “near zero valence” state of single Pd atom is obtained. Related description and explanation are supplemented in the revised manuscript.

[1] Nat. Commun. 2021, 12, 3783.

[2] Nat. Commun. 2021, 12, 302.

[3] Adv. Funct. Mater. 2021, 2108464.

[4] J. Phys. Chem. C 2020, 124, 24566.

[5] Angew. Chem. Int. Ed. 2021, 60, 16622.

[6] Mater. Today Energy 2021, 20, 100653

4) In DFT calculations, 0.58 eV of H₂O₂ decomposition barrier can also be progressed since a higher 0.81 eV of OOH formation on single Pd was also progressed. However, in the authors' test results, the catalyst with single Pd didn't decompose H₂O₂. Please explain this inconsistency between DFT and experimental results.

Response: Thanks for the comment. The chemical bond in O₂ is O=O double bond and that in H₂O₂ is O-O single bond. And the energy of the O=O double bond (about 498kJ/mol) is greater than that of the O-O single bond (about 131kJ/mol)). Therefore, it's reasonable that the calculated potential barrier for the formation of OOH (O₂→OOH 0.81eV) is higher than that

for the decomposition of H_2O_2 ($\text{H}_2\text{O}_2 \rightarrow 2\text{OH}$ 0.58eV). Since the reactants (O_2 and H_2O_2) and reactions are different, in our humble opinion, the comparison of its calculated barriers on one model (Pd_1/TiO_2) is not very necessary, and it might be more meaningful to compare the energy barriers of the same reaction on the different models such as Pd_1/TiO_2 and $\text{Pd}_8\text{O}_8/\text{TiO}_2$ in the manuscript.

5) The reviewer understands the difficulties of operando analysis. Although Flaherty group did the high pressurized operando analysis, the reviewer do not strongly recommend it this study. However, the key is which state dominated the catalytic activity, before the reaction (oxidized Pd) or after the reaction (metallic Pd)? It should be examined by reasonable evidences.

Response: Thanks for the constructive suggestions. This is a very good question. As we have responded to the Comment 3.2, the leading factor of catalytic activity may be attributed to the "near zero valence" single Pd atom derived from the *in-situ* catalytic reaction. In our humble opinion, it may be a dynamic process from the initial states to the "real-states" in catalysis, and we believe that the oxidation state of the as-prepared catalyst might be a necessary condition for reaching the "real-states" of the catalyst during the catalysis. However, we should clarify that the "real-states" of single Pd atom catalysts have not been well unveiled in the work, which might be the focus of future research. Related discussions are supplemented in the revised manuscript.

Comment 4. In DFT study section, it would be better to change "cluster" to "PdO cluster".

Response: Thanks for the comment. The "cluster" in DFT study section has been replaced with "PdO cluster".

Comment 5. The authors responded that XRD peak shift is not obvious, however only the 0.1%O-Pd/TiO₂ shows the shift as below. Since others are perfectly matched, it would provide information that 0.1%O-Pd/TiO₂ might have different lattice structure compared to others. Please reconsider this.

Response: Thanks for the comment. To verify the reviewer's concern, we have conducted XRD tests again and refined the results. We haven't observed the XRD peaks shift of samples this

time. Besides, the refined results show that the lattice parameters of TiO_2 between samples are almost identical (Supplementary Fig. 1c-f), so the effect of different loading on the lattice of TiO_2 could be excluded. The former observed XRD peak shift of 0.1%O-Pd/ TiO_2 might be caused by instrument problems or improper operation during the test. We have replaced the figure in the revised manuscript.

Supplementary Fig. 1c-f . The XRD pattern of different samples. (a) pure TiO_2 , (b) 0.1%O-Pd/ TiO_2 , (c) 1%O-Pd/ TiO_2 , (d) 3%O-Pd/ TiO_2 -450

REVIEWER COMMENTS

Reviewer #2 (Remarks to the Author):

I appreciate to the authors that they devoted themselves to making significant efforts for improving the quality of study and to providing constructive responses to the comments. However, the kernel of the questions is still not acceptable for me to recommend this study to Nature Communications. Detail comments are as below, and all comments are written based on the journal Nature Communications, please consider this point.

Comment 1. The relevance between “real catalytic active sites” and “the state that the authors investigated” is unclear and unacceptable. They stressed the single atoms catalyst has not been changed during the reaction, and the electronic state of the initial state of the catalyst (Pd^{2+}) is in accordance with that of DFT model. Since the authors claim the near zero valence state of catalyst could be the “real state”, then all characterizations including DFT couldn’t be rational for elucidating their catalytic activity.

- DFT model needs to be changed based on the electronic state of the palladium atom.
- They should provide observable single atoms by HAADF-STEM (without red circles, I cannot observe the atoms in this resolution). The used catalyst roughly shows 2 nm of agglomerated Pd particle behind the TiO_2 particle. It has to be clarified by EDS analysis including lower magnified image.
- EXAFS data for the used catalyst to examine the existence of a Pd-Pd bond.

Comment 2. Although the authors claimed the near 100% of selectivity for 2.5 hours, the explanation for the activity reduction in the recycle tests is insufficient for me to accept the situation.

Comment 3. The novelty and strong point of this study is insufficient to meet the high-level journal’s target. The approach of developing catalyst is a conventional and also similar with [P. Tian et al. / Journal of Catalysis 349 (2017) 30–40 37]. Moreover, below 4% of conversion under a high-pressure batch reactor is not such improved activity considering a number of reported papers. In a similar view, it is reported that near 0% of H_2O_2 decomposition can be achieved when using an extremely small amount of metal. Flaherty et al. reported at Science that 0.05 wt.% of Pd on SiO_2 also presented negligible H_2O_2 decomposition.

Reviewer 2

I appreciate to the authors that they devoted themselves to making significant efforts for improving the quality of study and to providing constructive responses to the comments. However, the kernel of the questions is still not acceptable for me to recommend this study to Nature Communications. Detail comments are as below, and all comments are written based on the journal Nature Communications, please consider this point.

Response: Once again, we thank the reviewer's constructive comments. We would like to address those comments as below.

Comment 1. The relevance between “real catalytic active sites” and “the state that the authors investigated” is unclear and unacceptable. They stressed the single atoms catalyst has not been changed during the reaction, and the electronic state of the initial state of the catalyst (Pd^{2+}) is in accordance with that of DFT model. Since the authors claim the near zero valence state of catalyst could be the “real state”, then all characterizations including DFT couldn't be rational for elucidating their catalytic activity. -DFT model needs to be changed based on the electronic state of the palladium atom.

Response: Thanks for the constructive comment. In our previous response, we have suggested that a near zero valence state of single Pd atom might be the active site. According to the reviewer comment, we have carried out some new experiment and re-examined the previous data, and drawn a different conclusion compared to our previous viewpoint. In a word, the Pd nanoparticle and single Pd atom are both observed in the used catalyst by using EDS mapping. Therefore, our previous viewpoint of near zero valence state of single Pd atom is inappropriate. Although the “real state” of catalyst remains unveiled due to the lack of operando spectroscopy technique, now we believe the initial oxidized state (Pd^{2+}) of single Pd atom mainly account for the high H_2O_2 yield and selectivity; thus, the DFT model in the manuscript is still close to the reality. The detailed results and explanations are showed in the following response. Related descriptions, discussions and conclusion are intensively revised in the re-submitted manuscript.

- They should provide observable single atoms by HAADF-STEM (without red circles, I cannot

observe the atoms in this resolution). The used catalyst roughly shows 2 nm of agglomerated Pd particle behind the TiO₂ particle. It has to be clarified by EDS analysis including lower magnified image.

Response: Thanks for the constructive comment. According to the suggestion, we tried our best to repeatedly test the STEM for the fresh and used 0.1%O-Pd/TiO₂; unfortunately, single Pd atoms were still not clearly observed, which might be the low z-contrast between Pd ($z = 46$) and Ti ($z = 22$) as well as the low palladium loading (0.1wt.%). For the used catalyst (Supplementary Figure 6), this time we have observed some aggregated Pd species (about 3 nm Pd nanoparticle) on the TiO₂ particle by using EDS mapping, however, the homogenous distribution of Pd species in other places suggest that many single Pd atom still existed. Then, we have re-examined and fitted the XPS results of used catalyst, and found the existence of oxidized Pd species other than the metallic Pd species (Supplementary Figure 7 & 8), which is consistent with STEM result. Thank you again for the valuable comment.

Supplementary Figure 6. The HAADF-STEM image of the used 0.1%O-Pd/TiO₂ catalysts.

Supplementary Figure 7. Pd 3d XPS spectra of 0.1%O-Pd/TiO₂ catalysts after 0.5 hours reactions.

Supplementary Figure 8. Pd 3d XPS spectra of 0.1% O-Pd/TiO₂ catalysts after 2.5 hours reactions.

- EXAFS data for the used catalyst to examine the existence of a Pd-Pd bond.

Response: Thanks for the constructive comment. We have observed the aggregated Pd species (about 3 nm Pd nanoparticles) by EDS mapping for the used catalyst, and the XPS results reveal the existence of metallic Pd species (Supplementary Figure 7 & 8, as below). Thus, the existence of a Pd-Pd bond could be confirmed. The use of EXAFS to verify Pd-Pd bond could be conducted for small Pd clusters and very useful; however, it might be not very necessary for Pd nanoparticles (about 3 nm) in our experiment.

Comment 2. Although the authors claimed the near 100% of selectivity for 2.5 hours, the explanation for the activity reduction in the recycle tests is insufficient for me to accept the situation.

Response: Thanks for the constructive comment. Since the metallic Pd species showed very poor H₂O₂ yield in the experiment (table 1, entry 9-11), the observed Pd aggregation (3 nm Pd nanoparticles) in the supplementary figure 6 might account for the loss of H₂O₂ yield in the recycle test. As to the near 100% of selectivity for 2.5 hours in Figure 3d, the conducted experiment is not cycle test; the initial state of catalyst used for every different time (from 0.5h to 2.5h) is always fresh, that is single Pd (2+) atom state. However, the initial state of catalyst used in the cycle test is changed in every round, and it's not the fresh catalyst anymore, which is revealed by our STEM & XPS characterization (Supplementary Figure 6 & 7 & 8). In our humble opinion, it might not be most appropriate to relate the H₂O₂ selectivity of non-cycle test with the H₂O₂ yield of cycle test. We have shown that the catalytic performance of initial

metallic Pd (table 1, entry 9-11) was far inferior to that of initial single Pd (2+) atom state. The decrease of H₂O₂ yield in the recycle experiment further indicates that the performance of the catalyst after reaction (co-existence of single Pd atom and Pd nanoparticles) is not as good as that of the fresh catalyst, highlighting the important role of the initial single Pd (2+) state. In a word, we believe the role of the initial single Pd (2+) atom state of catalysts is very important for both high H₂O₂ yield and selectivity in the experiment.

To further confirm this hypothesis, we have heated the used catalyst (after 5 cycles) at 350 °C for 3 h in the air, and the H₂O₂ yield increase from 74 mol/kg_{cat}/h to 100 mol/kg_{cat}/h, which is close to the fresh catalyst. It was also reported that the anneal treatment in the oxidative atmosphere will re-disperse the noble nanoparticle to the single noble metal atom on the oxide support. (ACS Catal. 2022, 12, 9, 4859–4871, J. Am. Chem. Soc. 2021, 143, 37, 15243–15249, Catal. Sci. Technol., 2020,10, 5772-5791). Thus, it's reasonable to believe that the recovered H₂O₂ yield might be caused by the recovery of single Pd (2+) atom through annealing in the air, which also suggest the anneal treatment in the air is an effective method to refresh the used catalyst. The result again confirms that the single Pd (2+) atom account for the high activity and selectivity towards the synthesis of H₂O₂. We have revised related descriptions and discussions in the re-submitted manuscript. Thank you again for the valuable comment.

Comment 3. The novelty and strong point of this study is insufficient to meet the high-level journal's target. The approach of developing catalyst is a conventional and also similar with [P. Tian et al. / Journal of Catalysis 349 (2017) 30–40]. Moreover, below 4% of conversion under a high-pressure batch reactor is not such improved activity considering a number of reported papers. In a similar view, it is reported that near 0% of H₂O₂ decomposition can be achieved when using an extremely small amount of metal. Flaherty et al. reported at Science that 0.05 wt.% of Pd on SiO₂ also presented negligible H₂O₂ decomposition.

Response: Thanks for the comment. We totally understand the reviewer's high standard for the journal of Nature Communications. However, this work is the first report on the direct synthesis of H₂O₂ by the use of single Pd atom catalyst which exhibited high activity (115 mol/g_{Pd}/h) and selectivity (>99%); besides, the H₂O₂ decomposition was shut down on the catalyst. As a result, a high H₂O₂ concentration of 1.07 wt.% is obtained in a batch, to the best of our knowledge, it

is the highest value reported so far. In our humble opinion, the work might promote the development of direct synthesis of H_2O_2 and exert an important influence on the subsequent catalyst design and mechanism research. Some detailed issues are addressed in the following: As to the approach of developing catalyst similar with another research [P. Tian et al. / *Journal of Catalysis* 349 (2017) 30–40]. First, the approach of developing catalyst is not point in the work; more importantly, P. Tian et al suggest the single Pd site catalyst is inactive for the direct synthesis of H_2O_2 in the mentioned reference, which is completely opposite to conclusion in our work.

As to the H_2 conversion rate, a similar issue had been raised by the reviewer in the first-time comment on the manuscript, and we had already given a response, which was affirmed and recognized by the reviewer in the second-time comment. Here, we have pasted the response again: “The reviewer carefully compared our data with Hutchings’ data and concluded that the H_2 conversion rate was low in our work. In our humble opinion, it may not be most appropriate to compare H_2 conversion between a small number of catalysts (2.5mg in our experiment) and a large number of catalysts (10mg in the Hutchings’ work); actually, we have investigated the effect of catalyst feeding (2.5, 5, 10 mg) on the catalytic performance in the original manuscript. (Fig.3a, 3b); for 10 mg 0.1% O-Pd/ TiO_2 catalyst feeding, the calculated H_2 conversion is 10.43% which is compared to the reviewer mentioned PdSn catalyst. (9% of H_2 conversion using 10 mg of 3 wt.% Pd-2 wt.% Sn/ TiO_2). Besides, the H_2O_2 yield and selectivity may be more commonly used to evaluate catalyst than the H_2 conversion.”

As to the issue of negligible H_2O_2 decomposition, we have carefully read the relevant literature, and don’t find the results about H_2O_2 decomposition in Flaherty's paper (*Science* 371, 626-632(2021)); in the supplementary materials of this paper, it was suggested that mass transfer can be accelerated when the Pd load is 0.05%. However, the H_2O_2 selectivity in this paper is less than 30%, so it’s reasonable to assume that H_2O_2 may decompose with 0.05 wt. % of Pd. For our single Pd atom catalyst, the experiments (Figure 3e, Supplementary Table 2-4) as well as DFT calculations (Figure 4f) confirm that the H_2O_2 decomposition hardly occur on the catalyst, which explained the accumulated high concentration of H_2O_2 (1.07%) well. The result is very meaningful for the decomposition of H_2O_2 can be avoided without adding any inhibitors or pretreatment of catalysts. The reviewer had raised a related decomposition issue in the first-

time comment to our manuscript, and we had already given a response.